# Preschoolers’ Developmental Profiles and School-Readiness in a Low-Income Canadian City: A Cross-Sectional Survey

**DOI:** 10.3390/ijerph17072529

**Published:** 2020-04-07

**Authors:** Chantal Camden, Léa Héguy, Megan Casoli, Mathieu Roy, Lisa Rivard, Jade Berbari, Mélanie Couture

**Affiliations:** 1School of Rehabilitation, Université de Sherbrooke, 2500 Boulevard de l’Université, Sherbrooke, QC J1K 2R1, Canada; megan.casoli@usherbrooke.ca (M.C.); melanie.m.couture@usherbrooke.ca (M.C.); 2Centre de recherche du Centre hospitalier universitaire de Sherbrooke, 12e Avenue N Porte 6, Sherbrooke, QC J1H 5N4, Canada; lea.heguy@usherbrooke.ca (L.H.); jade.berbari@usherbrooke.ca (J.B.); Elodie.Herault@USherbrooke.ca (P.R.T.); 3Centre intégré universitaire de santé et de services sociaux de l’Estrie—Centre hospitalier universitaire de Sherbrooke, 300 Rue King E, Sherbrooke, QC J1G 1B1, Canada; mathieu.roy7@usherbrooke.ca; 4School of Rehabilitation, McMaster University, 1280 Main St W, Hamilton, ON L8S 4L8, Canada; lrivard@mcmaster.ca

**Keywords:** child development, screening, early intervention, vulnerable populations, motor skills, community health services

## Abstract

A joint initiative between community and public health stakeholders in a low-income Canadian city was created to describe the developmental profiles of children aged 2–5 years. A two-phase, cross-sectional design was used. Children’s development was assessed using an online screening procedure. Those at risk of delays were invited for a school-readiness face-to-face brief assessment. Descriptive and exploratory analyses were conducted. In Phase 1, 223 families were screened; 100 children were at risk of delays (45%); 13% were at risk in ≥3 developmental domains; 26% were at risk in the fine motor domain. Risk of delay was associated with parental concerns, accessing more healthcare professionals, and using fewer public health/community programs. Lower incomes, and not attending day care showed trends towards an increased risk of delay. In Phase 2, 49 children were assessed; 69% were at risk of school-readiness delays; 22% had potential motor delays; 37% were at risk in the social domain. This study found a higher proportion of children at risk of delay than typically reported. Creating community partnerships could help identify all children needing developmental and school-readiness support. More research is needed to ensure these community-based partnerships are integrated into health/community programs responding to children’s needs and parental concerns.

## 1. Introduction

The importance of early childhood development and its impact on future health outcomes is well established [1,2], as are the effectiveness of early intervention programs when delays are present [3,4,5]. Identification of children demonstrating developmental delay who could benefit from early interventions programs is therefore crucial [6,7,8]. Pediatricians have a key role to play in identifying these children [9,10,11]. There are however many factors that may hinder pediatricians’ ability to observe and evaluate children’s development, including limited time and the physical environment of the medical office [12]. In addition, it is well recognized that parents know their children best and that their observations can be as reliable as those of health care professionals [13,14]. Identification of children at risk for delay should thus be a shared responsibility between pediatricians who could implement regular developmental surveillance and families who should be encouraged to voice their concerns and/or complete self-reported screening tools to identify potential delay.

There is also a growing recognition that other health care professionals and/or adults present in a child’s daily environment should be involved in identifying children who are at risk, especially educators [15,16]. Identification strategies adopted in the educational environment may help to identify children who might otherwise be missed by traditional surveillance and screening programs, since many families who are vulnerable might not have access to traditional health services [17,18]. Failure to reach most vulnerable populations is known as the inequality paradox [19]. Community initiatives, particularly in low-income communities, where a greater proportion of children may be at risk of delay [16,20], are needed to reach out to families and identify children who might also be at risk of school-readiness disadvantages.

The current study is a joint initiative between community organizations, public health clinicians and decision-makers, and health care researchers in a medium-sized Canadian city (Sherbrooke, in the Province of Quebec, Canada). This city has lower poverty indicators than the national average [21]. The region includes a mix of urban, semi-urban, and rural areas, has a population of approximately 500,000, and 93.4% is French speaking [21]. About half of this population lives in Sherbrooke, the 6th largest city in Quebec. Provincial statistics suggest that children in this region might be at greater risk of developmental delay than in other regions [21]. The research team wanted to strengthen partnerships to foster optimal development and support school transition, especially for children who are vulnerable. All families in the city were offered the opportunity for developmental screening for their children between the ages of 2 and 5, with community-based recruitment efforts focused in four low-income neighbourhoods.

The purpose of this study was to describe the development of children aged 2–5 years in a medium-sized low-income Canadian city. We sought to: (1) describe the children’s developmental profiles and school-readiness and (2) explore factors influencing the risk of developmental delay and school-readiness.

## 2. Materials and Methods

### 2.1. Design

A two-phase, cross-sectional study was designed in collaboration with partners following a participatory approach [22]. All subjects gave their informed consent for inclusion before they participated in the study. The study was conducted in accordance with the Declaration of Helsinki, and the protocol was approved on October 9, 2018 by the Ethics Committee of the CHUS research centre (project identification code 2019-2983).

### 2.2. Participants and Recruitment Procedures

Participants included children aged 27 months to 5.5 years and their parents. All participants were fluent in French or English (or with access to an interpreter) and lived in Sherbrooke, Quebec. Phase 1 recruitment was undertaken via traditional media (e.g., local radio station interview) and social media (e.g., Facebook page and targeted Facebook groups). Workshops about childhood development were also organized in kindergartens and community organizations in the four low socioeconomic neighbourhoods with attending families invited to participate in the study. Children determined to be at risk for developmental delay based on their Phase 1 screening tool scores were eligible to participate in Phase 2.

### 2.3. Data Collection Procedures and Instruments

#### 2.3.1. Phase 1

Data collection was undertaken entirely online. Parents were invited to complete a consent form and questionnaire online via the Research Electronic Data Capture (REDCap) [23], hosted securely on our university web server. Parents responded to questions about their child’s age and area of residence to ensure inclusion criteria were met. Additional data collection included questions about: (1) preferred contact information (n = 6); (2) family sociodemographics (n = 14); (3) use of educational, healthcare, and community services (n = 4); and (4) an age-specific version of the Ages and Stages Questionnaire Third Version (ASQ-3) [24].

The ASQ-3 is a valid and sensitive screening tool to assess children’s development, commonly used in health and social care settings [24]. Different aged-based ASQ-3 questionnaires are available for children 4 to 60 months to provide an overview of development. The ASQ-3 has five domains: communication, gross motor, fine motor, problem solving, and personal social skills. Questions differ according to the child’s age but the questionnaire asks respondents if the child is able to perform various activities. Options for responses are yes, no, and sometimes (e.g., Does your child catch a large ball with both hands? The 48 month questionnaire is available online at https://agesandstages.com/resource/asq-3-48-month-questionnaire/. For each of the five domains, scores are interpreted according to age-based cut-offs to determine children clearly at risk of delay, those who might be at risk and should be closely monitored, and those not at risk of developmental delay. All children determined to be clearly at risk or at risk only were eligible for Phase 2. The ASQ-3 also includes 5 yes/no questions asking parents if they have general concerns related to each of the developmental areas.

#### 2.3.2. Phase 2

Data collection was undertaken by a trained research assistant with a rehabilitation background, with the parent and child present. Data collected included: (1) the Developmental Indicators for the Assessment of Learning—Fourth Edition (DIAL-4) [25] and (2) the French-Canadian version of the Peabody Picture Vocabulary Test (PPVT) [26].

The DIAL-4 is a screening tool administered in person designed to identify children who may be in need of diagnostic assessment or early intervention across five developmental domains associated with school-readiness: Language, Motor, Concepts, Social, and Self-Help. In the United States (and increasingly in Canada), this test is used in systematic screening of children in preschool [27]. The DIAL-4 has a short administration time (30–45 min) and covers all developmental areas. However, as the DIAL-4 is unavailable in French, the Language domain could not be assessed, so this domain was replaced by the validated French-Canadian version of the PPVT. The PPVT tests the child’s receptive vocabulary and has good internal validity [28] and is sensitive for children from 2 years 6 months to 18 years [29].

### 2.4. Data Analysis

Descriptive analyses including means and frequencies were conducted on all study data using SPSS (version 25) (IBM Corp, New York, United States). ASQ-3 and DIAL-4/PPVT scores were interpreted based on cut-off scores to identify frequencies and percentages of children at risk of developmental delay in each domain. Parents responding yes to at least one of the five ASQ-3 general questions about developmental concerns were considered to have concerns. For a given ASQ-3 domain, if more than 2 questions were not completed, the domain score was not valid. If 1 or 2 domain questions were not completed, an average domain score was calculated from the completed items, as suggested by the test developers. For children to be included in the final analysis, they had to have valid scores for a minimum of 3 out of 5 domains. Chi-squared and Fisher’s exact tests (*p* = 0.05) were undertaken to compare scores of children participating in Phase 1 only to those participating in Phases 1 + 2 and to ascertain if sociodemographic and service utilization variables were associated with ASQ-3 or DIAL-4/PPVT outcomes (to estimate the effect size of the factors found to influence the risk of having a delay, we computed odds ratios).

## 3. Results

### 3.1. Participants

For Phase 1, 223 children were included; 100 of these were eligible for Phase 2, with 49 consenting to participate. Figure 1 describes the recruitment process. Table 1 presents the sociodemographics of family and child participants in Phase 1 only and Phases 1 + 2.

Table 2 describes parents’ developmental concerns (as measured by their responses to the ASQ-3 general questions) and their use of educational, health care, and community programs.

### 3.2. Phase 1

A high proportion (45%) of children were at risk of delay, in at least one domain, based on their ASQ-3 scores (Table 3). Thirteen percent (13%) of the children were deemed at risk of delay in three or more domains. Of the five ASQ-3 domains, the greatest percentage of children (27%) were found to be at risk of delay in the fine motor domain.

Being at risk in at least one ASQ-3 domain was associated with children being male (*p* < 0.01; OR = 2.44 (1.37,4.36)), parental concerns (*p* < 0.01; OR = 5.11 (2.86,9.12)), accessing a greater number of health care professionals (*p* = 0.001; OR = 2.54 (1.48,4.36)), and using fewer public health or community-based programs (*p* = 0.01; OR = 0.50 (0.29,0.85)). There was also a strong trend that having a lower family income increased the likelihood of a child being at risk of delay (*p* = 0.069). All other variables related to family sociodemographic characteristics and use of educational, health care, and community services presented in Table 1 and Table 2 were not associated with being at risk of delay.

### 3.3. Phase 2

With regards to characteristics of participants in Phase 2, there were significant differences between children and parents in Phase 1 only and those in Phase 1+2. There were more male children than female children participating in Phase 1+2 (*p* = 0.05; OR = 1.88 (0.95,3.70)); families reported more developmental concerns (*p* < 0.01; OR = 4.02 (1.93,8.37)), consulted a greater number of health care professionals (*p* = 0.02; OR = 2.08 (1.09,3.99)), and accessed fewer health and community programs (*p* = 0.02; OR = 0.49 (0.26,0.93)). No significant differences were observed for participants who were found to be at risk of delays, compared to those in Phase 1+2, in terms of sociodemographic characteristics and use of services.

Table 4 presents the findings for the 49 Phase 2 child participants. Sixty-nine percent of children were deemed at risk for delay related to school-readiness in at least one domain as per the DIAL-4/PPVT screening protocol. While potential motor skill delays were identified in 24.5% of the sample, the percentage of children at risk was the highest in the social domain (36.7%).

Being at risk in at least one domain of the DIAL-4/PPVT screening protocol was associated with children being male (*p* = 0.01; OR = 5.79 (1.54,21.79)) and with families consulting a greater number of health care professionals (*p* = 0.05; OR = 3.6 (1.01,12.81)). A trend was found for associations between being at risk and families with lower income (*p* = 0.06) and not attending a publicly funded day care (*p* = 0.07). All other variables related to family sociodemographic characteristics and use of educational, health care, and community services presented in Table 1 and Table 2 were not associated with being at risk of delay.

## 4. Discussion

The aim of this study was to describe the developmental profiles and school-readiness of children aged 2–5 years in a low-income Canadian city. In addition, we sought to explore factors influencing risk of both developmental delay and school-readiness.

We found a high proportion of children at risk, above the number generally reported by others [30], and consistent with studies reporting a trend between socioeconomic status and prevalence of developmental concerns and/or special care needs [2,17,21,30]. This highlights the need to focus identification efforts in lower socioeconomic neighbourhoods and to promote healthy child development in these communities. It is also possible that parents more concerned by their children’s development participated more in this study, generating a higher percentage of developmental delays. Yet it speaks about the importance of offering families opportunities for screening and consultation, to ensure they can voice their developmental concerns.

The association found between using fewer public health/community-based programs and greater risk of delay is surprising and of some concern. Typical assumptions are that children with delay access more health and community resources. Our results suggest the opposite. Access inequities and difficulties reaching out to families in vulnerable circumstances are well documented [17,18,31], but to our knowledge, this is the first time that program access is reported as a factor potentially influencing delay. This warrants more research but certainly suggests that not only paediatricians and family doctors might have a potential role in ensuring that families are aware of resources and supported in accessing services but also all community partners involved with preschool children.

Among children with developmental delay, a high proportion were identified as being at risk in Phase 2 and thus might be at risk of school-readiness delay. While we could not statistically explore this relationship, our findings and the literature suggest that children identified with developmental delay are at risk of facing more school challenges and may need to be supported in the transition to school. Developmental delays in skill acquisition are known to influence school-readiness, which includes the concepts of social and self-help skills. Since the underlying constructs assessed by the ASQ-3 and the DIAL-4/EVIPP are distinct as they relate to developmental delays and school-readiness respectively, it might explain why fine and gross motor skill issues were mostly identified in Phase 1, but only about a quarter of the children were identified as having motor issues in Phase 2. Self-help and social domains, which were the domains for which children were more at risk on the school-readiness tests, are broader concepts that might be influenced by a variety of developmental delays, including motor delays. It is also important to note that most children studied here were at risk in more than one domain for both developmental delay and school-readiness.

There are many implications of these findings for practice. Firstly, we would suggest the establishment of community partnerships designed to foster children’s development, especially in neighbourhoods where children are more at risk of developmental delays. These partnerships are crucial to reach out to the most vulnerable families, to identify children at risk of developmental delays, and to help them access support to optimize children’s development and school-readiness. Secondly, we would suggest the implementation of multimodal screening strategies within these partnerships. Online screening with validated tools such as the ASQ-3 offers the advantage of increasing the accessibility of the screening and might provide a cost-effective way to screen children. However, given the limitations in the responsiveness of the ASQ-3 or any test to identify all potential delays and the fact that all of families’ developmental concerns are worth taking into consideration, it is also important that families having concerns have access to health care providers. In addition to offering online screening, it is therefore also important to have community sites accessible where families can go and discuss their developmental concerns with a health professional. Likewise, including health care professionals in community teams to screen children and to help community workers recognize developmental delays would be an important strategy to ensure all families have access to developmental screening. Supporting community workers involved with the most vulnerable families in recognizing developmental delays would also be highly recommended. These hard-to-reach families may be less likely to use online screening or voice their concerns, yet children from these families are often at particular risk of developmental delays. Community workers could help vulnerable families access online screening and also raise awareness about the impact of developmental delays on future health and academic success.

Beyond screening, it would be important to ensure all children have access to activities promoting global development and early intervention as needed. Health and community-based programs fostering children’s development should ensure that program activities respond to children’s needs and thus foster children’s development in all spheres, including motor skills. Typically, health programs focus predominantly on health concerns or issues, and community programs more on psychosocial activities. It is possible that fostering development more globally, with health and community programs working together, may be more beneficial for children’s overall development. Ensuring that community programs focus on all areas of development, including motor skills, may be important and achieved by incorporating arts and crafts as fine motor activities as a way to develop skills and sport to enhance participation and fitness levels. Finally, ensuring that community initiatives supporting children’s development are well integrated into the continuum of care would ensure that children needing more support have access to early intervention as appropriate.

This study was designed in collaboration with health care and community organization partners, strengthening the relevance and implications of the findings to the local context. We particularly focused on children and families from low-income communities as a way of reaching out to families who may not have equitable access to resources and supports in their communities. Importantly, we explored the use of community resources to inform next steps for designing programs to match needs, including risk of delay and delayed school-readiness and the content of community programs to ensure that children and families’ needs are adequately addressed. However, we used a self-reported screening tool for identifying children at risk of developmental delay and our relatively small sample size limited exploration of associations between delay and school-readiness, specific age-related analyses in particular. The use of a language assessment for school-readiness focusing only on receptive language may have limited our findings. Finally, we experienced a high dropout rate, with less than half of the families where the child was deemed at risk of delay participating in Phase 2. This may have been related to parents not being interested in having their children fully assessed by health care professionals, especially knowing they would not get more support throughout this research even if developmental concerns were identified. Phase 2 also required greater travel and participation time than Phase 1, which could have discouraged families from completing the assessment. Yet, better understanding why some families having children at risk of developmental delays chose not to have their children assessed deserves more attention. Finally, despite high representation of the low-income neighbourhood in our sample, families with low incomes might be under-represented in our sample.

## 5. Conclusions

We found a higher proportion of children at risk of developmental/school-readiness delay than has been typically reported, with a trend towards an association between socioeconomic status and the prevalence of developmental concerns. This highlights a need to focus identification efforts on vulnerable populations and to support efforts aimed at promoting healthy child development in these communities. In particular, paediatricians may play a valuable role in ensuring that families are aware of resources and supported in accessing developmental services. We also identified factors influencing risk of delay including access to different types of kindergartens and use of health/community programs. The associations between parental concerns, developmental delay, school-readiness, and access to health/community programs warrant more research.

## Figures and Tables

**Figure 1 ijerph-17-02529-f001:**
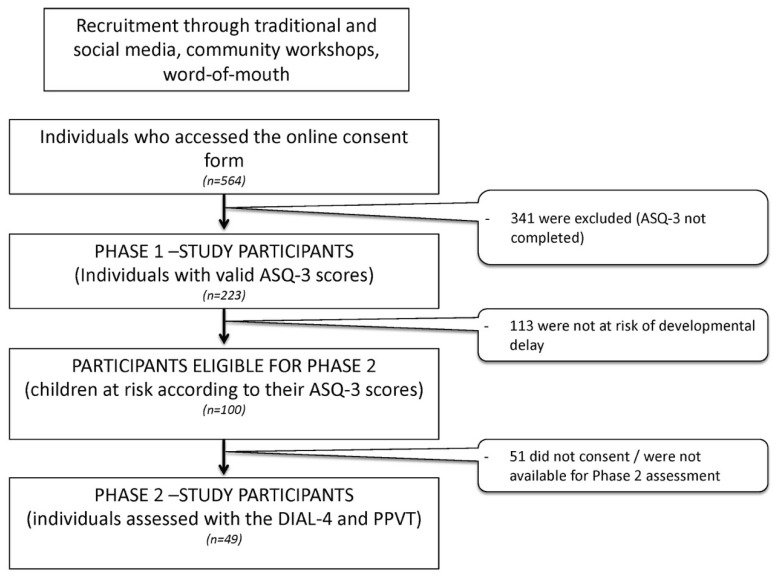
Flow chart of the recruitment process.

**Table 1 ijerph-17-02529-t001:** Parent and child sociodemographic information.

Characteristics	Phase 1 Only n = 174 (%)	Phases 1 + 2 n = 49 (%)
**Child sociodemographic information**		
*Child’s age (years)*		
2	37 (21.3)	18 (36.7)
3	74 (42.5)	13 (26.5)
4	43 (24.7)	12 (24.5)
5	20 (11.5)	6 (12.2)
*Child sex*		
Male	78 (52.3)	33 (67.3)
Female	71 (47.7)	16 (32.7)
*Child’s birth country*		
Canada	156 (91.8)	37 (84.1)
Other country	14 (8.2)	7 (15.9)
**Parents and families’ sociodemographic information**		
*Number of children in the family*		
1	36 (21.1)	8 (16.3)
2	98 (57.3)	32 (65.3)
3 or more	37 (21.6)	9 (18.4)
*Number of children between 2 and 5 years old in the family*		
1	130 (74.7)	34 (69.4)
2	44 (25.3)	15 (30.6)
*Residence in 1 of the 4 low socioeconomic neighbourhoods*		
Yes	21 (12.5)	6 (12.2)
No	147 (87.5)	43 (87.8)
*Relation to the child*		
Mother	152 (87.4)	43 (87.8)
Father	22 (12.6)	6 (12.2)
*Parent age (years)*		
18–30	48 (28.1)	11 (22.9)
31–40	109 (63.7)	33 (68.8)
41 or more	14 (8.2)	4 (8.3)
*Ethnicity*		
Caucasian	161 (94.2)	42 (87.5)
Other	10 (5.8)	6 (12.5)
*Civil status*		
Common law union	94 (55.0)	26 (54.2)
Married	56 (32.7)	15 (31.3)
Other	21 (12.3)	7 (14.5)
*Language*		
French	160 (93.6)	44 (91.7)
English	8 (4.7)	2 (4.2)
Other	3 (1.8)	2 (4.2)
*Education of the parent*		
Elementary school	5 (2.9)	0 (0.0)
High school	29 (17.0)	10 (20.8)
College	33 (19.3)	9 (18.8)
University	104 (60.8)	29 (60.4)
*Family income ($)*		
<10,000	3 (1.8)	1 (2.1)
10,000 to 29,999	17 (9.9)	2 (4.2)
30,000 to 59,999	24 (14.0)	8 (16.7)
60,000 to 79,999	25 (14.6)	10 (20.8)
80,000 to 99,999	29 (16.7)	8 (16.7)
100,000 to 149,999	56 (32.7)	15 (31.3)
>150,000	14 (8.2)	4 (8.3)
*Parent occupation*		
Working full time—salary	84 (48.3)	30 (61.2)
Working part time—salary	21 (12.1)	3 (6.1)
Independent worker	17 (9.8)	2 (4.1)
At home	16 (9.2)	2 (4.1)
Searching for a job	2 (1.1)	0 (0.0)
On family leave	15 (8.6)	5 (10.2)
Social welfare	5 (2.9)	1 (2.0)
Studying	15 (8.6)	4 (8.2)

**Table 2 ijerph-17-02529-t002:** Parents’ developmental concerns and reported use of educational, health, and community programs and services.

Variables	Phase 1 Only n = 174 (%)	Phases 1 + 2 n = 49 (%)
**Parents’ developmental concerns**		
*Parental concerns on the ASQ-3*		
Yes	80 (46.2)	38 (77.6)
No	93 (53.8)	11 (22.4)
**Parents’ reported use of programs and services**		
*Kindergarten type*		
Publicly-funded	113 (65.7)	30 (61.2)
Other	59 (34.3)	19 (38.8)
*Most frequent health care professional accessed*		
Doctor	158 (91.3)	44 (93.6)
Speech therapist	31 (17.8)	14 (28.6)
Physiotherapist	17 (9.8)	7 (14.3)
Occupational therapist	13 (7.5)	4 (8.2)
Psychoeducator	10 (5.7)	6 (12.2)
Social worker	5 (2.9)	6 (12.2)
Other	8 (4.6)	8 (16.4)
*Number of health and community services known*		
One or no program	13 (7.5)	6 (12.2)
Two or more programs	161 (92.5)	43 (87.8)
*Number of health and community services used*		
One or no program	72 (41.4)	29 (59.2)
Two or more programs	102 (58.6)	20 (40.8)

**Table 3 ijerph-17-02529-t003:** Percentage of children at risk of developmental delay based on their ASQ-3 scores (n = 223).

Risk of Developmental Delay Based on ASQ-3 Score	n (%)
Not at risk in any domain	123 (55.2)
At risk in at least one domain	100 (44.8)
In one domain	50 (22.4)
In two domains	21 (9.4)
In three or more domains	29 (13.0)
**Risk of developmental delay by ASQ-3 domain**	**n (%)**
Fine motor skills	59 (26.5)
Gross motor skills	39 (17.5)
Communication	36 (16.1)
Problem solving skills	35 (15.7)
Personal social skills	28 (12.6)

ASQ-3 = Ages and Stages Questionnaire Version 3.

**Table 4 ijerph-17-02529-t004:** Percentage of children at risk of school-readiness delay based on their Developmental Indicators for the Assessment of Learning—Fourth Edition/Peabody Picture Vocabulary Test (DIAL-4/PPVT) scores (n = 49).

School-Readiness Risk Category	n (%)
Not at risk	15 (30.6)
At risk in at least one domain	34 (69.4)
One domain	18 (36.7)
Two domains	9 (18.4)
Three domains	3 (6.1)
Four domains	3 (6.1)
**School-Readiness Risk by DIAL-4/PPVT domain**	**n (%)**
DIAL-4 Motor Skills	12 (24.5)
DIAL-4 Concepts	8 (16.3)
DIAL-4 Self-help	16 (32.7)
DIAL-4 Social	18 (36.7)
PPVT Communication	8 (16.3)

DIAL-4 = Developmental Indicators for the Assessment of Learning—Fourth Edition; PPVT = French Canada version of the Peabody Picture Vocabulary Test.

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
