# Peer review of "Preschoolers’ Developmental Profiles and School-Readiness in a Low-Income Canadian City: A Cross-Sectional Survey"

_ijerph, 2020, doi:10.3390/ijerph17072529_

Round 1
Reviewer 1 Report
This is an interesting paper that takes a two step approach to evaluating and identifying children at risk of developmental delay and difficulties with school readiness. This is an interesting and important topic that this paper addresses well. However, I do have several concerns that, if addressed, would likely strengthen the paper substantially.
First, could an analysis of those screen positive at stage 1 but did not go onto complete stage 2 be done? A comparison based on the baseline data between these two subgroups would identify if there were any systematic variations in those who choose to continue the study and those who choose not to continue the study. It would be interesting to know if the families whose children were identified as having multiple problems in different domains were either more or less likely to engage in the stage 2 study. It might also be interesting if there were no differences.
The discussion should include more about the practical implications of this work. What kind of implications are there from this study about how best to identify these individuals? Were parental concerns a good indicator of their ratings of concerns on the ASQ? Or was it only the ASQ that parents could register their concerns? Could identification be done on the basis of parental concerns more generally, without using this measure? Could the screening measure be included in routine care in order to best identify the children that may benefit from community interventions? It doesn’t appear as if the authors have fully evaluated the implications of the screening being broadly successful in identifying children at risk of having problems with school readiness.
Another issues, which is more minor, is that the authors should indicate the full statistics when reporting these. I noted that the statistics conducted were mentioned, but full statistical information for each reported results should be presented. A measure of effect size would also be useful, particularly when trying to interpret marginally significant results.
The abstract needs further work and focus. It would be good to include the overall context and rational of the study at the start. Overall it is dense and difficult to follow. Are statistics really needed here, or can you have a higher-level discussion of the results and implications of the results? Also use of acronyms without definitions should be avoided.
Minor comments:
Line 59 – it is unclear what “region” is being discussed here? Is this Sherbrooke or Quebec? You later mention the greater territory of Sherbrooke – is that what was being referred to here? More specifics needed.
Line 189 – typo – “ensure they can voice” not “ensure the can voiced”
Author Response
Dear Reviewer 1
Thank you very much for reviewing our manuscript. We found your suggestions and comments very helpful and have made revisions to the manuscript accordingly. Modifications made as per your suggestions (bold-face type) as well as Reviewer 2’s suggestions, are highlighted in the text using track changes, and are also detailed in the attached cover letter.

Reviewer 2 Report
This article addressed a rather important issue, i.e. “Preschoolers’ developmental profiles and school-readiness in a low-income Canadian city”. The participatory design of the study, which includes family members and researchers from different sections makes it quite interesting.
The article is well-structured, well-written and easy to read. The rationale supports the importance of the study while the methodology is clearly explained. The well-presented results were followed by a decent discussion.
However, I identified some minor issues, so I suggest that the authors address them.
In lines 96-104 the tool ASQ-3 is described.
An exemplar depicting a part of this tool (of one of the 5 domains of ASQ-3) would be useful for the reader. It could be added at an appendix
Lines 202-204.
In my view, this is not a valid interpretation or needs more elaboration as motor skills were also assessed in phase 2, but a lower percentage was revealed. So here, a question emerges. How do you explain the parents' concern of fine-motor skills and the evidence of higher developmental delay of social skills in phase 2? Please elaborate more on this.
Lines 228-235, about the high dropout rate in Phase 2.
The authors might add some propositions of what community services can do to convince parents to participate in such evaluations, since, as we know parents often do not accept that their child may have some developmental delay.
Lines 235-236. (“…families with low incomes might be underrepresented in our sample”)
The authors should elaborate more on this issue. This might be associated with the way Phase 1 is realized. Do all low-income families have access to digital tools of communication? Do they are acquainted with the usage of such platforms or digital resources (i.e.. to complete a questionnaire on-line)?
Author Response
Dear Reviewer 2
Thank you very much for reviewing our manuscript. We found your suggestions and comments very helpful and have made revisions to the manuscript accordingly. Modifications made as per your suggestions (bold-face type) as well as Reviewer 1’s suggestions, are highlighted in the text using track changes, and are also detailed in the attached response letter.
